# TRPM8 thermosensation in poikilotherms mediates both skin colour and locomotor performance responses to cold temperature

Hannan R. Malik[1], Gabriel E. Bertolesi [1✉] & Sarah McFarlane [1✉]

Thermoregulation is a homeostatic process to maintain an organism's internal temperature within a physiological range compatible with life. In poikilotherms, body temperature fluctuates with that of the environment, with both physiological and behavioral responses employed to modify body temperature. Changing skin colour/reflectance and locomotor activity are both well-recognized temperature regulatory mechanisms, but little is known of the participating thermosensor/s. We find that *Xenopus laevis* tadpoles put in the cold exhibit a temperature-dependent, systemic, and rapid melanosome aggregation in melanophores, which lightens the skin. Cooling also induces a reduction in the locomotor performance. To identify the cold-sensor, we focus on *transient receptor potential* (*trp*) channel genes from a Trpm family. mRNAs for several Trpms are present in *Xenopus* tails, and Trpm8 protein is present in skin melanophores. Temperature-induced melanosome aggregation is mimicked by the Trpm8 agonist menthol (WS12) and blocked by a Trpm8 antagonist. The degree of skin lightening induced by cooling is correlated with locomotor performance, and both responses are rapidly regulated in a dose-dependent and correlated manner by the WS12 Trpm8 agonist. We propose that TRPM8 serves as a cool thermosensor in poikilotherms that helps coordinate skin lightening and behavioural locomotor performance as adaptive thermoregulatory responses to cold.

[1] Hotchkiss Brain Institute and Alberta Children's Hospital Research Institute, Department of Cell Biology and Anatomy, University of Calgary, Calgary, AB, Canada. ✉email: gbertole@ucalgary.ca; smcfarla@ucalgary.ca

Thermoregulation is an important homeostatic process that occurs in organisms to maintain internal temperature within physiological ranges. There are differences, however, in the ways that organisms thermoregulate. Homeotherms, like birds and mammals, maintain their body temperatures over a wide range of environmental temperatures[1]. Feathers and fur in their integument were selected during evolution as insulation systems that contribute to homeostasis[1]. In contrast, the temperature of poikilotherms fluctuates with that of their environment, with poikilotherms primarily employing physiological and behavioral responses to keep the body temperature within a physiological range compatible with life[2–4]. The integument of poikilotherms is exposed, and altering skin colour/reflectance is considered an important physiological process that contributes to thermoregulation[3,5,6]. Poikilotherms also use behavioural responses to thermoregulate; basking[7,8], shuttling between environments[9], burrowing underground to different levels[10], altering body position to adjust heat exchange[11], and altering locomotor performance[12–14]. The thermosensation mechanisms in poikilotherms that underlie temperature-dependent skin colour change and behavioural responses are poorly understood, and the extent to which the mechanisms overlap for the two processes is unclear. Such knowledge is of critical interest for understanding thermoregulation in poikilotherms.

In poikilotherms, rapid skin colour changes are produced by the movement of coloured or reflective pigment organelles within the dermal chromatophore cells[15,16]. The physiological mechanisms that underlie how temperature change triggers the aggregation/dispersion of pigment-containing organelles to alter skin color are unknown. The capacity for absorbed visible light to be converted into heat led to the suggestion that lightening or darkening the skin relates to cooling or warming of body temperature, respectively[3,17]. Support for this idea, known as the "thermal melanism hypothesis", comes from observations in certain poikilotherms that dark-colored individuals warm up faster than their light-colored counterparts[6,15,16,18,19]. Changes in skin colour/reflectance, however, underlie ultraviolet (UV) protection and cryptic coloration (i.e. background mimicry)[3,17,20,21], as well as thermoregulation, and all three mechanisms are influenced by the energy spectrum (ea.: UV, visible and infrared lights). Thus, determining the physiological mechanisms that specifically underlie thermoregulation aspects of skin pigmentation are complicated by the fact that in nature a key driver of temperature change is light, but light changes skin pigmentation for additional benefits to the organism. For instance, in amphibians pigmentation cycles between a dark state during the warm days and a light skin during the cold nights[22–24], potentially related to UV protection. Thus, the determination of the physiological mechanisms that regulate skin colour change for thermoregulation requires a focus only on the variable of temperature, something that has yet to happen.

Thermoregulation depends on cation channels of the superfamily of proteins known as transient receptor potential (TRP) channels[25]. TRP channels are classified into two groups, and further into seven subfamilies based on protein homology. Group 1 contains the thermosensory families: TRPM, TRPA, TRPV and TRPC[25,26]. TRP channels are characterized by six transmembrane domains with intracellular C and N termini, and are sensitive to a broad range of stimuli, which include temperature, pH, osmolarity and chemicals[25,26]. The role of TRP channels as skin thermosensors to adjust the internal temperature of homeotherms is well known[27,28], as are their roles in regulating the differentiation, apoptosis, and melanogenesis of mammalian melanocytes[29]. TRP channel subfamilies are conserved evolutionarily at the molecular level[26,30], but the temperature ranges of channel activity between taxonomic clade can differ[26,30,31].

For example, members of the TRPM and TRPA subfamilies in mammals sense cool temperatures, while in amphibians and reptiles the TrpA1 channel acts as a heat sensor[31–33]. TRP channels do participate in behavioural responses to temperature in poikilotherms. Trpa1 in sensory nerve fibers of the pit organ of snakes acts as a heat sensor to detect infrared signals critical for hunting[34,35]. Trpv1 and Trpm8 are the hot and cold sensors, respectively, that coordinate the shuttling of crocodiles between cool and hot environments to maintain body temperature[9]. Trp channels may also play a role in long-term adaptive mechanisms to environmental temperature. For instance, Trpm8-induced calcium activity in motorneurons of *Xenopus laevis* embryos raised in a cold environment promotes their survival, which translates to enhanced locomotor performance in the cold[12].

In addition to their participation in behavioural mechanisms, we hypothesised that in poikilotherms Trpm channels directly and rapidly regulate pigment aggregation/dispersion induced by changes in environmental temperature. To test this hypothesis, we used the frog *Xenopus laevis* to analyze the effect of cold temperature on skin pigmentation, identified Trpm channels expressed by skin pigment cells, and used pharmacological approaches to identify a role for Trpm8 as a temperature sensor for skin colour change. Interestingly, we find Trpm8 is also associated with a cold-mediated slowing of locomotor performance, arguing that two key thermoregulatory mechanisms share a common photosensor to help to maintain temperature homeostasis.

## Results

### Cold temperature induces aggregation of melanosomes in skin melanophores independent of melatonin released from the pineal complex.

We first asked if *Xenopus* tadpoles change their skin pigmentation in response to a change in temperature, focusing on the response to cold. Of note, skin pigmentation in *Xenopus* tadpoles relies only on melanophores, which contain the pigment melanin in organelles called melanosomes. Dispersion or aggregation of the melanosomes produces darkening or lightening of the skin, respectively[22–24]. We used a temperature change as stimulus and analyzed alterations in skin pigmentation, by measuring the pigmentation index in the dorsal head of stage 42/43 *Xenopus* tadpoles as described previously[36]. Specifically, stage 42/43 tadpoles were switched from 24 °C to 6 °C, a temperature experienced in nature by *Xenopus* in the winter months (Fig. 1a, b). Because *Xenopus* skin colour changes with the day/night cycle, as well as with the brightness of the background surface (background adaptation)[21,24,37], it was important to have temperature be the only variable to change. Therefore, the analysis of pigmentation variation by temperature was always performed at the approximate middle of the light phase and with constant overhead illumination and surface colour (Fig. 2a). Switching the tadpoles from 24 °C to 6 °C induced a significant aggregation of melanosomes within 20–30 min, resulting in skin lightening as seen by a reduced pigmentation index (Fig. 2b). The lightening persisted for hours (Fig. 2b) and was systemic in that it involved melanophores of the head, belly, and tail (Fig. 2c, d).

The fast kinetics and the systemic response observed with cold-induced skin lightening are similar to those described previously for tadpoles switched from light to dark conditions[24]. In the latter scenario, melatonin released from the pineal complex in the dark phase induces systemic melanosome aggregation. Because in nature "dark" involves both temperature and light variables, and temperature regulates melatonin synthesis and release by zebrafish pinealocytes[38], we hypothesized that lightening triggered by cold temperature is mediated by melatonin release from the pineal complex. However, cold-mediated melanosome

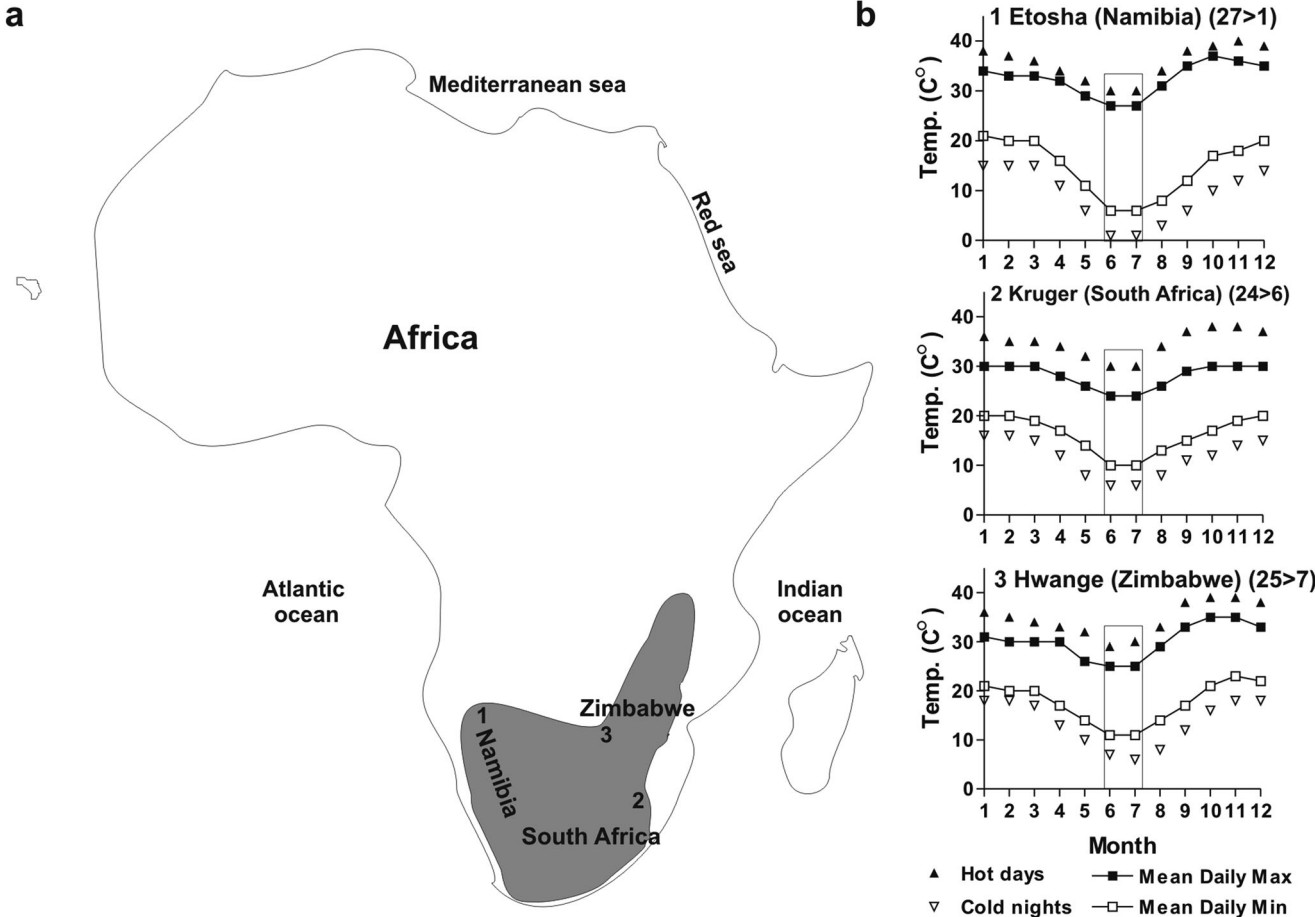

**Fig. 1 Monthly temperature variation in the approximate natural habitat of *Xenopus laevis* in Southern Africa. a** Map of Africa indicating approximately the native range of *Xenopus laevis* (modified from Furman et al., 2015) and three natural parks (numbers) located in this region. **b** Mean temperature data taken from the indicated natural parks including warm [hot days (filled triangles) and mean daily maximum (filled squares)] and cold conditions [cold nights (empty triangles) and mean daily minimum (empty squares)]. The mean temperature variation in June and July was used to devise a cooling paradigm (24 °C to 6 °C) that mimics the temperatures experienced by *Xenopus* in their native environment during the coldest months of winter (rectangle).

aggregation occurred in pinealectomized tadpoles (Supplementary Fig. 1), ruling out this hypothesis. Further, the fact that 30 min cooling (24 °C to 6 °C) produced a pigmentation index intermediate to that triggered by dark conditions at 24 °C (Fig. 2c, d), suggests that the pigmentation response to dark is mediated by both temperature and light variables.

Our cooling paradigm used a sudden and defined change in water temperature. But tadpoles would normally experience gradual changes in temperature, as water acts as a buffer with temperature change occurring slowly. To detect how the pigmentation index of tadpoles adjusts to gradual decreases of temperature we designed an experiment based on the feature that different volumes of water produce distinct temperature decay kinetics upon cooling, allowing us to slowly cool the embryos' environment at different rates. To run these experiments, tadpoles were placed in either 100 or 500 ml of salt solution at 24 °C and then moved to 6 °C, and we measured the pigmentation index and the temperature of the solution over time. The final pigmentation index of tadpoles moved directly from a 24 °C to 6 °C solution (Fig. 2a) was similar to those that gradually reached 6 °C, and independent of the container volume (Fig. 2e). These data indicate that with an abrupt 24 °C to 6 °C temperature switch the kinetics of the skin pigment change reflect mainly those of melanosome movement, while with the large volume containers the kinetics reflect the gradual decrease in temperature. Yet, the final pigmentation colour mirrored a

common final 6 °C temperature, rather than the method by which this temperature was reached (Fig. 2e). Of note, the skin of embryos in the 100 ml container exhibited more rapid lightening, reflecting the fact that the temperature of the water cooled more quickly than with the larger volume (Fig. 2e, *$p$ < 0.05 at 30 and 45 min). Interestingly, the plots of temperature versus pigmentation index show a similar sigmoidal function for the two fixed volume containers (Fig. 2e, insert), suggesting that the aggregation induced by cooling occurs gradually in a manner linked to the temperature.

We next analyzed if a scenario existed whereby melanosome dispersion occurred at cool temperatures. We compared the pigmentation response induced by cold for dark-adapted tadpoles (highly aggregated pigment) to that of tadpoles adapted to a black background (highly dispersed pigment) (Fig. 2f). Dark-adapted tadpoles at 24 °C dispersed melanosomes when switched to a black background with light shone from above (24 °C), with a kinetic that required between 4 to 6 h to reach the maximum dispersion[24] (Fig. 2f; compare dark 24 °C moved to black background at 24 °C vs control, constant black background at 24 °C). When dark-adapted tadpoles were switched to a black background surface at 6 °C, the pigmentation index increased over time, suggesting melanosome dispersion had occurred. However, melanosome dispersion never reached that observed for tadpoles placed on a black background at 24 °C (Fig. 2f; compare dark 24 °C to black background at 6 °C vs dark 24 °C to black

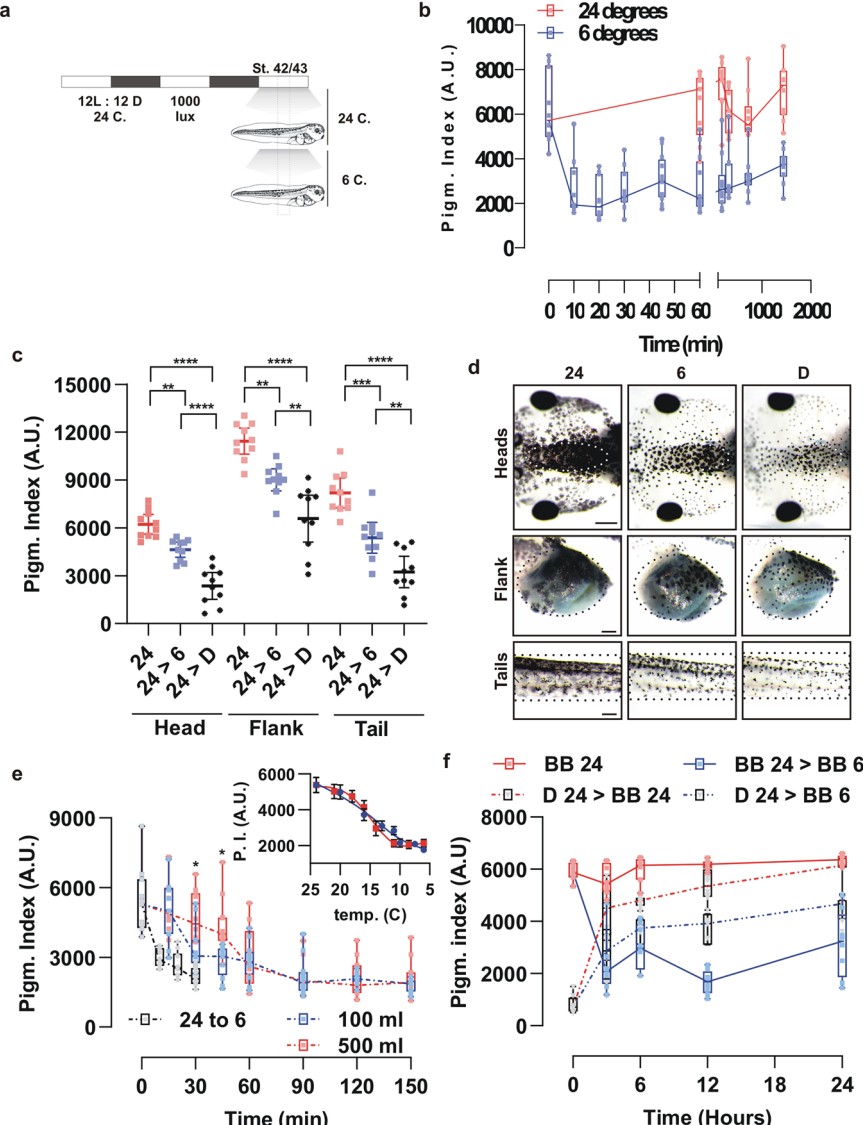

**Fig. 2 Colour change induced by cooling is a systemic response mediated by a fast and temperature-dependent movement of melanosomes in skin melanophores. a** Schematic representation of the experimental approach. Embryos were raised on a 12 h ON/12 h OFF cycle until stage 42/43 with light shining from above (1000 lux). Embryos were maintained at 24 °C for at least two days prior to reaching stage 42/43. Analysis of the effect of temperature on skin pigmentation was performed at approximately the middle of the light phase cycle (rectangle), by switching to bathing media at 6 °C, while keeping constant the overhead light and surface colour. **b** Pigmentation index quantified from digital images of the dorsal head of *Xenopus* embryos at different times after being moved from 24 °C (red dots and lines) to 6 °C degrees (blue dots and lines) [individual data points (*n* = 9 embryos) and a box plot (25th to 75th percentile) are represented; *N* = 3 independent experiments]. **c, d** Comparative pigmentation index (**c**) and pictures (**d**) of the dorsal head, lateral belly (flank) and the tail of tadpoles at 24 °C (red dots) or 6 °C (blue dots), or switched to dark (black dots) conditions at 24 °C (D; dark) for 30 min (horizontal bar represented the mean ± 95% CI; *n* = 10 embryos; *N* = 3 independent experiments). Scale bar = 0.2 mm. **e** Head pigmentation index of tadpoles in 100 ml (blue dots and dashed line) or 500 ml (red dots and dashed line) containers at 24 °C moved to a 6 °C incubator. The solution temperature was monitored over time and is plotted vs. the pigmentation index (P.I.); blue: 100 ml; red: 500 ml; (Insert). Tadpoles switched from a 24 to 6 °C solution, as shown in 2 A, are also indicated (24 to 6; black dots and dashed line) [individual data points (*n* = 10 embryos) and box plot (25th to 75th percentile); *N* = 3 independent experiments]. **f** Temperature response (24 °C -red- to 6 -blue- °C) over time as quantified by the head pigmentation index of tadpoles previously adapted to either a black background (BB; solid lines) (darker embryos; melanosome dispersed) or dark conditions (D; dashed lines) (lighter embryos; melanosome aggregated). [individual data points (*n* = 8 embryos) and box plot (25th to 75th percentile); *N* = 2 independent experiments); **p* < 0.01, ****p* < 0.001; *****p* < 0.0001; ANOVA followed by Bonferroni's test. **p* < 0.05 *t*-test.

background at 24 °C). These data suggest that the skin pigmentation achieved at a set cold temperature is a physiological response that is reached independently of the initial pigmentation level (melanosome aggregation or dispersion from black-surfaced or dark-adapted tadpoles, respectively). Aggregated melanosomes, however, is the final state in a colder environment under identical light conditions.

In summary, cooling induces a systemic lightening of the skin independently of the pineal complex. The level of pigmentation is temperature dependent and adjusts to both abrupt and gradual temperature variation. The final skin colour achieved with movement to a specific temperature is constant, and does not depend on the pigmentation level of the embryo prior to movement.

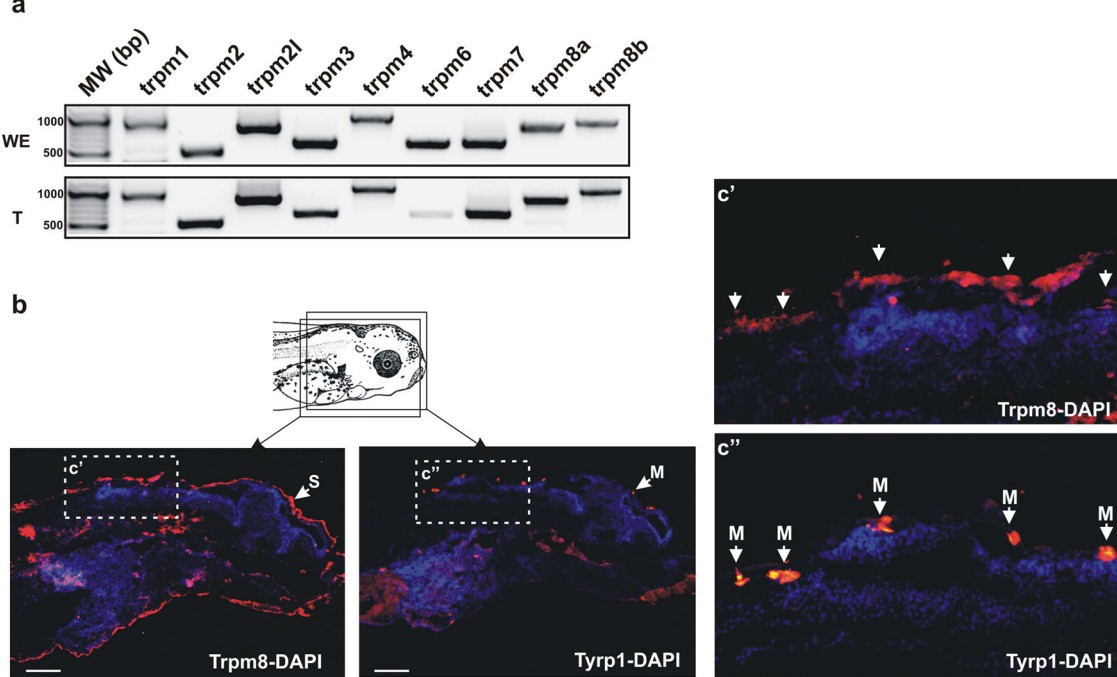

**Fig. 3 *trpm* channel mRNA and TRPM8 protein expression by skin melanophores. a** Representative RT-PCR analysis of mRNA expression for *trpm1-trpm8b* in the whole embryo and tail at stage 42/43 ($N = 3$ independent samples). *trpm6* mRNA was detected slightly in 1 of 3 independent replicates. **b** Immunohistochemistry against TRPM8 and Tyrosinase related protein 1 (Tyrp-1) in consecutive sections (schematic). DAPI staining (blue) was used to visualize cell nuclei and facilitate overlapping of adjacent sections. Boxed areas are shown enlarged (c′ and c″). Immunolabel displays co-localization of Tyrp-1 (arrows) in melanophores (M) of the skin (S) with Trpm8. Scale bar = 100 μm.

**Identification of *trpm* channel genes in *X. laevis* and expression of Trpm8 by skin melanophores.** Our hypothesis was that Trp channels mediated the rapid skin pigmentation changes observed with cooling temperature. We focused on the TRPM subfamily involved in cold sensation in poikilotherms, though we identified in *X. laevis* all of the thermosensory *trp* channels (Supplementary Fig. 2). We first determined *trpm* mRNA expression in the tail of stage 42/43 tadpoles by RT-PCR. mRNA was present for all *trpm* genes in the whole embryo and isolated tails, with the exception of *trpm6* in the tails (expressed at low levels in 1 of 3 experiments) (Fig. 3a and Supplementary Fig. 3).

We were particularly interested in Trpm8, in that the pigmented melanophores of *Xenopus* are related evolutionary to mammalian melanocytes[39], and both human melanocytes and melanoma cells express TRPM8[29,40,41]. Further, TRPM8 in the epidermis mediates sensory responses to local skin cooling in mouse[42] and humans[43]. We asked whether Trpm8 protein was expressed by skin melanophores. To confirm Trpm8 expression in melanophores in vivo we performed immunohistochemistry on adjacent sections for Trpm8 and a melanophore/melanocyte marker, Tyrosinase-related protein 1 (Tyrp-1)[44]. Trpm8 was detected throughout the epidermis, including in Tyrp1-positive melanophores [head, (Fig. 3b)]. Of note, adjacent sections were used for single labeling in that the TRPM8 and Tyrp1 antibodies were both produced in rabbit. In support of Trpm8 expression by skin melanophores, Trpm8 was expressed by a stable melanophore cell line generated from stage 35 tadpoles (MEX cells)[45] (Supplementary Fig. 4a)[46]. Of note, we found MEX cells aggregated their melanosomes when the temperature was dropped from 24 to 6 °C (Supplementary Fig. 4b, c). Thus, a potential candidate thermosensor for inducing melanosome movement mediated by cooling, TRPM8, is expressed by skin melanophores in vivo and in vitro.

**Melanosome aggregation induced by cooling likely requires Trpm8.** To determine whether Trpa1 or Trpm8 channels participate in cold-mediated melanosome aggregation we asked if pharmacological agonists could mimic the melanosome aggregation induced by cold when administered to tadpoles at 24 °C. Of note, the sensitivity of TRPA1 and TRPM8 channels to their respective agonists, AITC and menthol, is conserved[31,34,35]. AITC activates TRPA1 channels in both *Xenopus laevis*[32] and rodents[35], while menthol (WS12 is the active compound of menthol) induces ionic currents upon binding of *Xenopus* and mammalian TRPM8[26,33,35]. As expected, AITC had no effect on skin pigmentation at any of the tested concentrations (Fig. 4a). In contrast, a dose-dependent skin lightening was induced by the TRPM8 agonist WS12 (Fig. 4a). The aggregation induced by WS12 was reversible, with the pigmentation index reaching similar values to 24 °C controls 6 h after wash out of the compound (Fig. 4a). The specific involvement of Trpm8 in the cold-induced aggregation was supported by using a TRPM8 antagonist, PF05105679, which blocks cold-induced behaviours in humans and guinea pigs[47,48]. PF05105679 inhibited cold-induced aggregation in a dose-dependent manner, but had no effect when added at 24 °C (Fig. 4b). Further, a TRPM4 blocker, CBA, that has no significant activity against TRPM8[49,50], did not affect the lightening induced by cooling at any of the tested concentrations (Fig. 4c). Together with the WS12 agonist data, these results point strongly to Trpm8 as the thermosensor involved in the skin lightening induced by cold.

**Skin lightening induced by cold temperatures correlates with locomotor performance.** Trpm8 participates in an adaptive developmental response that occurs over days, whereby *trpm8* + motor neuron survival is enhanced when *Xenopus* embryos are reared at cold temperatures[12]. Whether motor

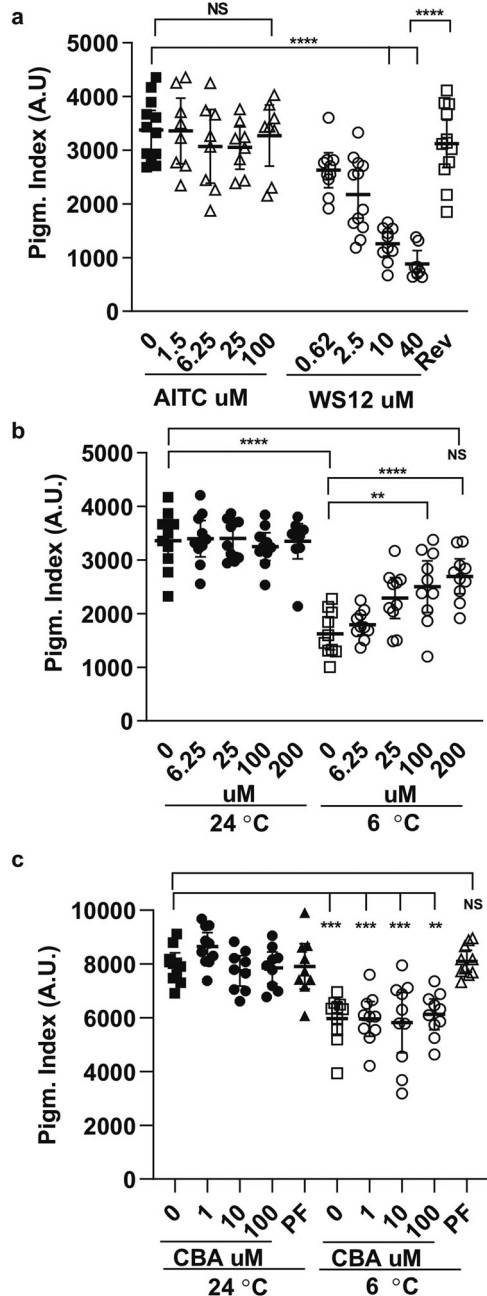

**Fig. 4 Melanosome aggregation response to Trp channel agonists and antagonists. a** Pigmentation index for stage 42/43 *Xenopus* tadpoles treated at 24 °C for 30 min at the indicated concentrations of TRPA1 (AITC) and TRPM8 (menthol; WS12) agonists. Reversibility (Rev) 6 h after removing the WS12. **b, c** A TRPM8 antagonist (PF05105679) blocks melanosome aggregation induced by cool temperature (6 °C) (**b**), while the TRPM4 blocker CBA has no effect (**c**); Each dot represented the measurement in one tadpole, and the bar is the mean with 95% confidence interval. $n = 9$ embryos; $N = 3$ independent experiments; **$p < 0.01$, ***$p < 0.001$; ****$p < 0.0001$; ANOVA followed by Bonferroni's test.

activity is regulated on a more immediate time scale by Trpm8 is unknown. Here, we directly compared the short-term effects of temperature on skin pigmentation and locomotor performance at an individual tadpole level; We asked if the distance individual tadpoles swam over 30 min was correlated with their level of skin pigmentation at 24 °C and 6 °C. To avoid activating skin mechanoreceptors we did not use the touch-induced movement

assay employed in a previous study with *Xenopus* tadpoles[12], and instead analysed "free" locomotor performance. We initially employed stage 42/43 tadpoles for these experiments, as this was the stage we used to analyse the effect of temperature on skin pigmentation. In the absence of inducing stimulus, however, stage 42/43 tadpoles alone in a dish were largely motionless; the average swim distance over 30 min (Fig. 5a) was almost zero (Fig. 5b, c). Thus, the unstimulated motor behaviour of isolated stage 42/43 tadpoles could not be assessed. Instead, free swimming responses required older larvae (stage 45/46), as is the case for two other behaviours: visual avoidance (stage 45/46)[51] and background preference (stage 44/45)[52,53]. Indeed, the locomotor performance at stage 45/46 differed significantly between tadpoles at 24 °C and 6 °C (Fig. 5b, c). At the end of the movement assay, the pigmentation index of each individual tadpole was determined. Similar to what we found with younger tadpoles (stage 42/43), the skin was lighter at 6 °C than at 24 °C (Fig. 5d). Interestingly, the level of pigmentation and the locomotor performance were correlated (Fig. 5e), with the more sedate tadpoles having a lighter skin.

We next analyzed if a similar correlated response occurred with tadpoles treated with the TRPM8 agonist, WS12. The locomotor response of tadpoles placed at 24 °C was reduced significantly by WS12 in a dose-dependent manner (Fig. 5f). Additionally, melanosome aggregation and lightening of the skin of stage 45/46 tadpoles were induced by WS12 (Fig. 5g), as we showed earlier with the younger tadpoles. The tadpoles at the 40 μM WS12 dose showed a narrow range of motor performance, with most tadpoles being inactive (Fig. 5h). This narrow range made it difficult to use this group to determine if locomotor performance and skin pigmentation were correlated. At 5 μM WS12, however, motor performance was more variable, and we found that the skin pigmentation responses were positively correlated ($p = 0.018$), with less active tadpoles having lighter skin. Of note, a significant positive association was observed between skin pigmentation and locomotor performance when tadpoles across all treatment groups were included in the analysis ($p < 0.001$). Together, these results show a positive correlation between locomotor performance and skin pigmentation, suggesting both processes share a common mechanistic sensor, possibly Trpm8.

## Discussion

Physiological change of skin pigmentation and behaviour responses are two common mechanisms used by poikilotherms to maintain body temperature at a physiological range compatible with life. Here, we identify a single molecular mechanism associated with both adaptive mechanisms. Our results show that in response to an ethologically-relevant cold temperature, a lighter skin is achieved via the aggregation of melanosomes of skin melanophores. Melanophores throughout the skin show this response, in a manner independent of the pineal complex (melatonin). The identification of mRNAs of Trpm family members in the tail, and a pharmacological approach with agonist (menthol) and antagonist (PF05105679) of TRPM8, suggest that Trpm8 serves in skin melanophores as the thermosensor of cold-mediated skin colour change. Finally, our data indicate that temperature-mediated changes in skin pigmentation and locomotor performance are associated phenotypes, linked to a common thermosensor, Trpm8.

The physiology of melanism is complex and influenced by physical (e.g., temperature and light), adaptive (e.g., background adaptation), and social (e.g., mating) responses. Thus, in our experiments it was important that the only variable we changed was temperature, keeping light conditions from above and below (surface) constant. Further, we based our cold temperature on the

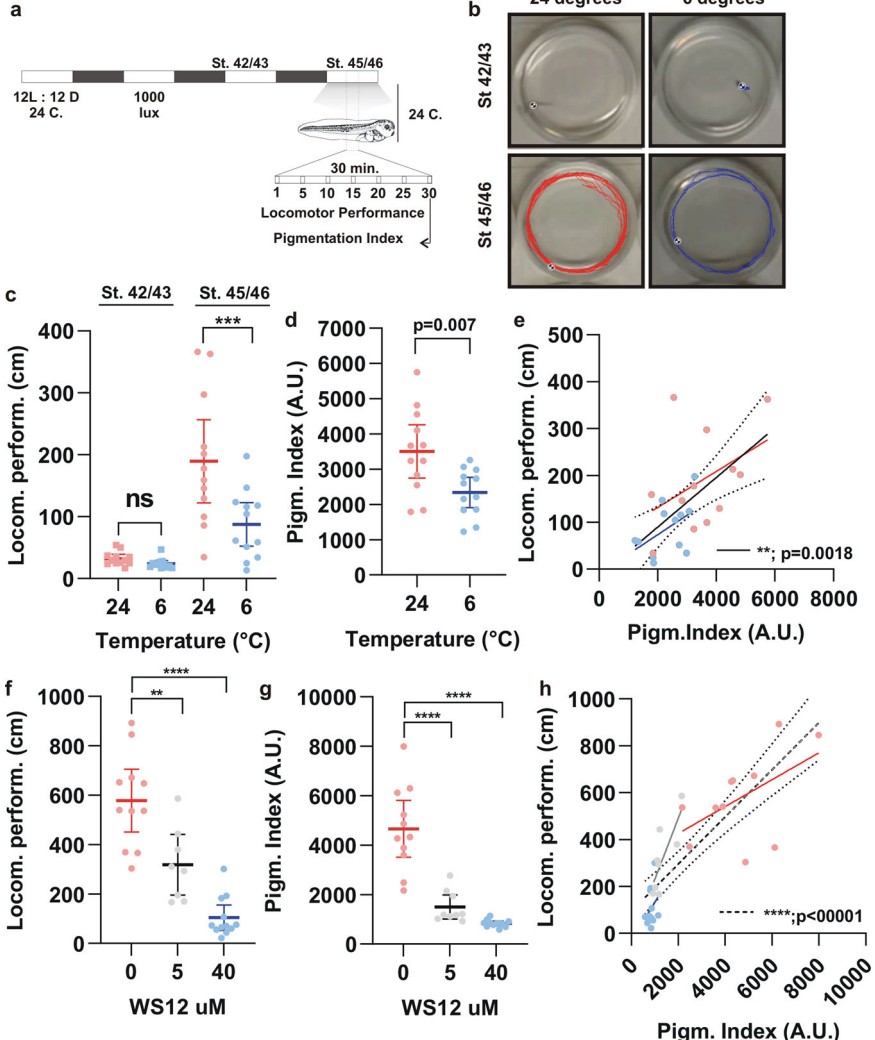

**Fig. 5 Skin pigmentation and locomotor performance. a** Schematic representation of the experimental approach. The locomotor performance was analyzed for stage 45/46 embryos by recording swim activity (1 min) every 5 min for 30 min. The pigmentation index of each tadpole was determined at the end of the study. **b** A representative analysis of the locomotor performance determined by tracking the distance swum at 24 °C (red) and 6 °C (blue). **c**, **f** Locomotor performance of embryos at 24 °C (red dots) or at 6 °C (blue dots) (**c**) or at 24 °C in the presence of WS12 (without WS12: red dots; 5 µM: grey dots; 40 µM: blue dots) (**f**). **d**, **g** Pigmentation index of stage 45/46 embryos at 24 °C or at 6 °C (**d**) or at 24 °C in the presence of WS12 (**g**). **e**, **h** Correlation between locomotor performance and the pigmentation index. The linear regression for each treatment is indicated with a red (24 °C) or blue (6 °C) solid line. Dotted lines represent the 95% confidence interval while the dashed line is the Pearson correlation coefficient (two-tail) from all data. **$p < 0.01$, ***$p < 0.001$; ****$p < 0.0001$; ANOVA followed by Bonferroni's test or Student's *t*-test in d. Each dot represents the measurement for one tadpole, and the bar is the mean with 95% confidence interval; $n = 8$ embryos; $N = 2$ independent experiments.

temperature records from the last 30 years from three National parks located in southern Africa where *Xenopus laevis* are native. Aggregation of melanosomes in this paradigm is not a noxious response to an abrupt temperature change, in that an identical alteration in skin pigmentation occurs when the water temperature is cooled slowly. Additionally, the extent of melanosome aggregation adjusts to the final temperature level, independent of whether pigment granules are initially aggregated or dispersed. Temperature appears to be sensed directly by melanophores, in that melanosomes also aggregate when isolated cultured melanophores are cooled.

The fact that the Trpm8 agonist (menthol; WS12) and antagonist (PF05105679) produce opposite effects on melanosome distribution argues strongly for Trpm8 as the main cold thermosensor in *Xenopus* melanophores. The electrophysiological properties of *Xenopus laevis* Trpm8[54] correlate with the

pigmentation response induced by temperature, enforcing the idea that Trpm8 is the main thermosensor involved in melanosome aggregation. For instance, Trpm8 is almost inactive at 24 °C (upper part of the sigmoid curve; Fig. 2e insert) and the half-maximum thermal activation current (EC50 13.89 °C) occurs at approximately 14 °C, which agrees with the sigmoid curve for the variation in skin pigmentation with temperature (EC50 ~14 °C; Fig. 2e, insert). mRNAs for other Trpm channels are present in the skin, possibly in melanophores, and further investigation is necessary to understand their potential roles in melanism.

Data from the *Xenopus* poikilotherm suggest an evolutionarily conserved role for Trpm8 in vertebrate skin thermosensation. The thermal activation response for the channel is almost identical between *Xenopus laevis* and *X. tropicalis* Trpm8[54], while the EC50 for the thermal activation of Trpa1, the heat sensor, is significantly lower in *X. laevis* than in *X. tropicalis*[33]. These Trp

channel properties likely reflect the optimal temperatures to which the two species have adapted[55]. Indeed, the two species show dependence of temperature for the locomotor performance;[13,56]. The optimal temperature range for motor activity in *X. laevis* is 16–22 °C and for *X. tropicalis* is 22–28 °C[57]. TRPM8 as a cold sensor also appears conserved in mammals, in that the TRPM8 mouse knockout is insensitive to cold temperatures, but can perceive heat and mechanical stimuli[42], and in humans, TRPM8 associates with sensory responses to local skin cooling[43]. In mammals, activation of TRPM8 in melanocytes, either by cooling or with the agonist menthol, reduces cell viability and decreases pigment synthesis in vivo and in vitro[29,41]. In *X. laevis* the activation of Trpm8 in melanophores aggregates melanosomes. Thus, despite distinct Trpm8-mediated responses, skin lightening is the end result of Trpm8 activation in both mammals and poikilotherms. Of note, the thermosensation role of TRPA1 is not conserved in that the channel acts as a heat sensor in amphibians[33] but as a cold sensor in mammals.

The thermal melanism hypothesis argues that darkening the skin is beneficial when temperatures drop, since more light is absorbed and converted to heat. Indeed, corporal heating of darker animals is common for terrestrial reptiles like snakes[58,59] and lizards[3,19,60–62]. The melanosome aggregation we observe with cold temperatures does not support this hypothesis, since colder conditions lighten rather than darken the skin. Amphibians are generally not thought to use basking behaviour for thermoregulatory purposes[7]. Instead, for aquatic organisms, such as tadpoles (our results) or semi-aquatic reptiles (turtle)[63], melanism generally decreases when the temperature drops, as long as conditions of surface and environmental light are maintained. Such a mechanism may also be present in adult amphibians, in that lighter-skinned wild frogs exhibit higher body temperatures than their darker counterparts[7].

Two plausible explanations for why amphibian skin lightens in the cold involve skin permeability and skin pigmentation serving as a mediator of UV protection. With the former, the skin of amphibians is highly permeable, which produces high rates of evaporative water loss[7,10]. In one species, *Bokermannohyla alvarengai*, where the relationship between evaporative water loss, body temperature and pigmentation colour was investigated, skin darkening and increased evaporative loss were associated with a higher temperature, with evaporative loss potentially serving as a physiological mechanism to lower body temperature[7,64]. Whether the skin colour change contributes to the process, however, is unclear. UV protection is another possible explanation for why we find melanism decreases with a temperature drop. Lighter skin is more susceptible to the harm of UV light, and in fish and frogs UV exposure induces melanosome dispersion[65–68]. In nature, rapid temperature drops and a reduction in UV exposure are associated with the darkness of night. The responses to temperature and light may have become linked over time, in that both cold and dark trigger a physiological aggregation of melanosomes. In agreement, the change in skin pigmentation produced by dark appears more robust than that we observe in response to a cold temperature alone (Fig. 2c, f and ref. [7]).

Our data argue that Trpm8 mediates both a behavioural response to temperature and a physiological skin pigmentation response. Indeed, we find both responses are temperature sensitive and exhibit a comparable dose-dependent response to WS12, suggesting strongly that the same thermosensor underlies the two processes. Which cells participate still needs to be determined. Trpm8 in melanophores likely triggers the pigmentation response. Additionally, Trpm8 could alter secretion of extrinsic regulators from nerves, such as the neurotransmitters acetylcholine, adrenaline and noradrenaline[69], to regulate skin pigmentation. Trpm8 immunolabel throughout the epidermis could also indicate the participation of neuronal fibers in the skin as mediators of temperature-dependent changes in locomotor activity. Of note, spinal motor neurons also express Trmp8[12]. While it is possible that low temperatures affect locomotor performance in a Trpm8-independent manner through alterations in the physiological properties of skeletal muscle cells, the fact that the Trmp8 agonist produces a similar slowing of embryo movement as cold suggests that Trpm8 thermosensation is involved. Many amphibian species move between terrestrial and aquatic environments as an additional thermoregulatory tool. The highest night time corporal body temperatures are registered for pond-residing adult frogs, with the water acting as a temperature buffer during the nocturnal drop in air temperature[70]. Interestingly, a study in three extant marine reptiles shows that melanism may have contributed to their ability to exploit cold environments[71], arguing for an evolutionary association between melanism and cold thermoregulation in aquatic animals.

In our study, we measured the temperature-dependence of both movement and melanism responses over a short 30-minute period. Rearing *Xenopus laevis* embryos at a cold temperature over days results in enhanced survival of Trpm8-expressing motorneurons in the embryonic spinal cord[12]. As a result, cold-reared embryos show greater locomotion in response to touch than their warm-reared siblings when both groups are placed in cold water. Thus, Trpm8 appears to function as a thermosensor in both short- and long- term (developmental time; days) adaptations of motor behaviour in response to being placed in a cold environment. Together, these data may explain why tadpole species living in environments with lower fluctuations of temperature and mean temperatures show a lower temperature for optimal swimming and more narrow thermal tolerance ranges[72]. Trpm8 appears as a critical sensor in organisms for thermal sensitivity and thermal adaptation, and through regulation of changes in skin pigmentation and the locomotor performance likely influences survival fitness under cold conditions.

## Methods

**Embryos, drug treatment and cool response.** *Xenopus laevis* tadpoles show robust and quantifiable responses to skin pigmentation[22,52] and were thus a model of choice to understand the role of skin pigmentation in the coordination of a response to cold temperature. The Animal Care and Use Committee, University of Calgary, approved procedures involving frogs and embryos which was signed by Dr. Derrick Rancourt (AC21-0148). Adult *Xenopus laevis* were obtained from NASCO (Wisconsin, USA). Embryos were obtained by induced egg production from chorionic gonadotrophin (Intervet Canada Ltd.) injected females and in vitro fertilization according to the standard procedures. Embryos were staged according to Nieuwkoop and Faber using Xenbase (http://www.xenbase.org) RRID:SCR_003280. Embryos were maintained at 16 °C until stage 24/26 (48 h), and then reared at 22–24 °C until stage 42/43 in Marc's modified Ringer's (MMR) solution (100 mm NaCl, 2 mm KCl, 2 mm CaCl$_2$, 1 mm MgCl$_2$, 5 mm HEPES pH 7.4), under light cycles of 12 h ON/ 12 h OFF (light = 1000 lux or approximately $1.5 \times 10$–4 W/cm$^2$).

The temperature experiments required a non-noxious cold temperature. We chose 6 °C based on the average temperatures recorded over the last 30 years, obtained from the Meteoblue website (https://www.meteoblue.com/en/weather/ historyclimate/), of three national parks in southern Africa (Etosha/Namibia; Kriger/South Africa and Hawange/Zimbawe); Regions to which *Xenopus laevis* are native[55] (Fig. 1A, B). To provide a temperature stimulus, embryos were set in petri dishes in two identical chambers (40 cm length × 20 cm width × 25 cm height) that contained two independently-powered parallel T5 bulbs (F875 cool white fluorescent; light output, 470 lumens, colour temperature, 4000; colour rendering index, 60) residing in a 24 °C or a 6 °C incubator. Chambers contained a white surface unless specifically mentioned and the petri dishes were covered with aluminum foil for experiments requiring dark conditions. For immunohistochemistry experiments, embryos at 48 h post fertilization were treated with 0.02% 1-phenyl-2-thiourea (PTU), an inhibitor of eumelanin pigment formation[36]. AITC (Allyl isothiocyanate; oil of mustard; Sigma Aldrich), WS12 [Menthol; (1 R*,2 S*)-N-(4-Methoxyphenyl)-5-methyl-2(1methylethyl)cyclohexanecarboxamide)], PF05105679 [3-[[[(1 R)-1-(4-Fluorophenyl)ethyl](3-quinolinylcarbonyl)amino]methyl]benzoic acid] and CBA [4-Chloro-2-[[2-(2-chlorophenoxy)acetyl]amino]benzoic acid] (all from Tocris Bioscience, Burlington, ON, Canada) were added to the rearing solution at the indicated concentrations. For each drug, tadpoles in the same solvent as the drug were

used as control. After treatment, tadpoles were fixed with 4% paraformaldehyde for pigmentation assessment.

**Assessment of pigmentation index**. We quantified changes in skin pigmentation by measuring skin pigmentation indices as described previously[36]. Briefly, pictures of the dorsal head of tadpoles were taken using a stereoscope (Stemi SV11; Carl Zeiss Canada, Ltd., Toronto, Canada) and a camera (Zeiss; Axiocam HRC), with identical conditions of light, exposure time and diaphragm aperture. Pictures were converted to binary white/black images using NIH ImageJ (U. S. National Institutes of Health, Bethesda, MD) public domain software. The density of positive pixels was measured, and the statistical significance ($p < 0.05$) between experimental groups was determined with GraphPad Prism 9 by using multiple ANOVA followed by Bonferroni's *post hoc* test or *t*-test ($n \geq 8$; $N \geq 3$).

**Screening of TRP channels in X. laevis, protein alignment and phylogenetic analysis**. In mammals, TRPA and TRPM are the subfamilies that sense cool temperatures[26]. Here, we focused on TRPM channels as TRPA1 serves as a heat sensor in amphibians and reptiles[31–33]. To identify all Trpm members in *X. laevis* we performed a screen using the sequences from a variety of organisms with a curated protein reference sequence. The gene sequences were used to blast the *X. laevis* genome with a cut-off expected (E) value for positive EST candidates set at 0.05. In Supplementary Table 1 we list all the Trpm channel genes identified and the NCBI accession numbers for their sequence. Phylogenetic analysis of the predicted amino acid sequences obtained from the genes, aligned using multiple sequence alignment (MUSCLE), was used to build a hidden Markov model (HMM) by using a maximum-likelihood architecture construction algorithm. Phylogenetic analysis and alignments (Supplementary Fig. 2) were performed by using the public domain MEGA X version 10.04 software (https://www.megasoftware.net/)[73].

**RT-PCR and cloning**. To determine if *trpm* channels are expressed by skin melanophores we assessed mRNA expression by RT-PCR. Total RNA was obtained from whole embryos (positive control) and isolated tails (cut below the cloaca to separate the tail and upper body) using TRIzol (Invitrogen) according to the manufacturer's protocol. Isolated tails were analyzed as a substitute for the skin, given the absence of organs and specialized tissues within the tails relative to the whole embryo. Single-strand cDNA was produced from RNA samples (5 μg) by priming with oligo(dT) primers using SuperScriptTM IV reverse transcriptase (Invitrogen) according to the manufacturer's instructions. All PCR amplifications were carried out in a total volume of 20 μL with 1 μL of cDNA, 2 μL of primers, 7 μL of water and 10 μL of 2X PCR master mix (Thermo Scientific, IL). PCR amplifications were carried out at 45 cycles and with an annealing temperature of 55 °C. PCR products obtained from cDNA were cloned into TOPO-pCRII (Invitrogen) vectors and sequenced to confirm identity. The cloned sequences were submitted to GenBank-NCBI (Supplementary Table 1).

**Immunohistochemistry for Trpm8**. To verify that Trpm8 protein was expressed by skin melanophores we used an anti-human TRPM8 (rabbit polyclonal; 1/1000 dilution; antibodies-online.com). Alignment revealed that the human and *Xenopus* TRPM8 protein sequences were similar in the region used to generate the human TRPM8 antibody (Supplementary Fig. 5A). To verify the specificity of the antibody for *Xenopus* Trpm8 we performed a Western Blot assay on protein isolated from stage 43 *Xenopus laevis* tadpoles. Protein was measured with bicinchoninic acid using a BCM protein assay kit (Thermo Scientific, IL). Sixty μg of protein/lane was separated on a 10% polyacrylamide gel and then transferred to a polyvinylidene difluoride membrane (Bio-Rad), and immunoblotted with anti-human TRPM8. Specific peroxidase-conjugated secondary antibodies were utilized to detect proteins by enhanced chemiluminescence (PerkinElmer Life Sciences). The antibody recognized two bands with a similar molecular weight (~130/150 kDa) to that reported previously for human TRPM8[74] (Supplementary Fig. 5B).

Immunohistochemistry with anti-TRPM8 was performed on cryostat sections of stage 42/43 embryos using standard procedures. Fixed samples of *Xenopus* embryos were cryoprotected via washing with PBT (PBS; 0.1% Triton; 0.05% BSA), followed by immersion in an RNAse-free PBS solution containing 30% sucrose for 30–45 min. Samples were then embedded in Optimal Cutting Temperature medium (OCT; Tissue TEK), frozen at −80 °C until sectioning, and 12 μm transverse sections collected with a Leica CM 3050 S. Standard procedures were performed on slides for immunohistochemistry using primary antibodies [rabbit polyclonal anti-human TRPM8 (1:200 dilution; antibodies-online.com, ABIN351227) and rabbit polyclonal against human Tyrp-1 (1:200 dilution; Thermo Scientific, IL; PA5-81909)] and a secondary antibody (1:1000 dilution of Alexa Fluor 488). Nuclei were stained with DAPI (1 μg/μl).

**Melanophore (MEX) cells culture**. To test if *Xenopus* skin melanophores show a direct pigmentation response to temperature we employed the melanophore (MEX) cell line originally generated from stage 35 *Xenopus laevis* embryos[75]. Cells were maintained in growth medium (70% Leibovitz's L15 medium with 25% added water and supplemented with 5% fetal bovine serum (Invitrogen)) without antibiotics. Cells were maintained at room temperature. Cells were adapted to 24 °C for

24 h before cooling to 6 °C. During the cooling paradigm, light shining from above (1000 lux) was maintained during all culture conditions, as described previously. Cells were fixed with 4% paraformaldehyde and stained with DAPI (1 μg/μl) before imaging.

**Locomotor performance**. Two species from the *Xenopus* genus, *X. tropicalis* and *X. laevis*, show alterations in their locomotor response to temperature changes[13,56]. To determine if the rapid changes in locomotory response to cooling required Trpm8 channel function we performed temperature-dependent movement assays. To assess tadpole movement single tadpoles were set in 35 mm dishes containing MMR solution at 24 °C or 6 °C (9–12 dishes on a tray) and assessed concurrently by using a Logitech C920S camera. Trays were kept in either the 24 °C or 6 °C incubator, with light shining from above. Both incubators were adjacent to the behaviour recording device, and trays with dishes were removed every 5 min (0–30 min) to record the movement (total distance swam) for one minute. In this manner, the temperature of the MMR in each dish did not change during the 30 min of the study. At the end of the experiment, each tadpole was fixed for assessment of skin pigmentation, and the recordings were used to determine their locomotor performance. The free online software, Kinovea version 0.8.15, was used to track the movement of each tadpole.

**Microscopy**. Images of embryos were taken with an Axio-Cam HRc (Carl Zeiss) on the Stemi SVII stereomicroscope (Carl Zeiss). Section images were processed for brightness and contrast with Adobe Photoshop 7.0.

**Statistics and reproducibility**. Statistical analysis included either a Student's *t*-test or an ANOVA followed by Bonferroni's test. Significance was considered at $p < 0.05$. GraphPad Prism 9.0 software was used for statistical analysis of data and graphic preparations. Experiments were performed three times ($N = 3$) unless indicated. Since pigmentation index varies between hatches, a representative experiment is show in each figure. The independent experiments showed similar trends. Experimental treatments contained a minimum of 8 tadpoles ($n$; generally, $\geq 9$) which are represented in the figures by data points. In addition, figures show a box plot (25th to 75th percentile) or the mean and 95% confidence interval or standard deviation of the mean. Immunohistochemical analysis were performed at less three times ($N = 3$) containing 4 independent tadpoles ($n = 4$) in each experiment. CorelDraw 10.0 was used to compile multipaneled figures.

**Reporting summary**. Further information on research design is available in the Nature Portfolio Reporting Summary linked to this article.

## Data availability

Data have been deposited in the Dryad repository site with an assigned unique digital object identifier (DOI); https://doi.org/10.5061/dryad.6q573n61f. Cloned sequences were deposited in Gene Bank with access numbers provided in Supplementary Table 1.

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

## Acknowledgements
This work was supported by a Discovery Grant from the Natural Sciences and Engineering Research Council of Canada (NSERC) to S.M., and an NSERC Studentship to H.R.M. We thank Carrie Hehr and Nilakshi Debnath for excellent technical assistance and to Dr. Validimir Rodionov at the University of Connecticut for providing the MEX cell line.

## Author contributions
H.R.M. and G.E.B. designed, performed, and analyzed the data of the experiments. G.E.B. and S.M. supervised the study. G.E.B. and S.M. wrote the paper and critically revised the manuscript.

## Competing interests
The authors declare no competing interests.
