## [Peer Review File · Communications Biology]

Reviewers' comments:

Reviewer #1 (Remarks to the Author):

Malik H. R., Bertolesi G. E., McFarlane S. "TRPM8 thermosensation in ectotherms mediates both skin colour and locomotor performance responses to cold temperature"

The authors studied the effect of cooling on skin pigmentation and motor activity, considering these reactions as thermoregulatory in poikilothermic animals - the object of the study was *Xenopus laevis* embryos and tadpoles. An attempt has been made to consider the molecular mechanisms of these processes and to determine the role of TRP ion channels in these processes. The results obtained allow to conclude that the TRPM8 ion channel is involved in changes in both skin color and locomotor activity under the action of cold. Experiments with the TRPM8 ion channel agonist and antagonist quite convincing evidence of this.

There are some questions with work.

1. In the title and introduction, the authors use the terms ectotherms, exotherms, but exotherms include poikilothermic animals and plants. It is better to use the terms poikilotherms and homeotherms, which indicate the features of maintaining body temperature in animals. Plants were not studied.
2. The number of individuals is not indicated everywhere.
3. The authors use a variety of methods to solve the problem, which certainly adorns the article. However, Screening of TRP channels in *Xenopus laevis* appears to be redundant for the intended purposes, since the focus of the study was on cold-sensing channels, and since *Xenopus laevis* TRPA1 appears to function as a heat-sensing channel, TRPM8 remains. Possibly, TRPM2 from the TRPM family also can be checked.
4. It remains not entirely clear whether TRPM8 is involved in the slowing down of motor activity, it may be due simply to a decrease in muscle temperature and this process goes in parallel with the activation of TRPM8. This conclusion can be formulated more accurately.
5. A change in skin pigmentation, namely, its lightening, may be associated not with thermoregulation, but with mimicry, because in cold water, transparency is greater (many protozoa disappear, the mobility of many living beings decreases).
6. In general, a lot of work has been done with a variety of methods; an interesting result has been obtained on the participation of the cold-sensitive ion channel TRPM8 in the regulation of skin pigmentation, another functional significance of this ion channel.

Reviewer #2 (Remarks to the Author):

Dear Authors,

I have read the manuscript entitled "TRPM8 thermosensation in ectotherms mediates both skin colour and locomotor performance responses to cold temperature" submitted to Biology Communications. My overall impression of the work is that this type of study combining behavioral test with phenotypic with experiments at mechanism levels, useful to understand proximate and ultimate causes of colour change (intensity of colouration to be specific), is highly valuable; but I have some major concerns regarding writing, the lack of clear hypothesis, and experimental design.

In general terms, the study is interesting. Anyway, I believe that it needs a lot of work because it does not seem to be hypothesis-motivated, without clear hypothesis/questions.

Introduction and discussion are well written but I didn't find a proper connection to Methods and Results sections. Method and Results should be rewritten. There is no mention of experimental design in the Method section making it difficult to understand why you performed the analyses. This Section is a group of analysis descriptions without arguments. I recommend reading the manuscript considering the audience perspective to understand the experience of the reader (a lot of details appear late in the manuscript that would be much easier if they appear earlier). A lot of

details are missing such as sample size, origin of specimens, treatments description, arguments for choosing treatments temperature, etc. Results section possess sentences that are NOT results, some sentences redundant with introduction, some discussions and sentences that are clearly Methodology. Also, the Results section presents new questions that should be part of preliminary studies or complete different manuscripts (for example, analyses on black background adaptation?). Finally, the discussion is well written but some paragraphs should be avoided because they are not related to this study (melanism as UV protective). Some of these aspects appeared also in the Introduction section and also could be shortened. The theoretical framework presents some critical points to discuss. I think there is a confusion about causes/consequences in the relationship between locomotion-temperature and I don't think the melanism hypothesis for thermoregulation process is the best framework for this aquatic phase. Moreover, the work is over dimensioned considering a family of TRP channels when they focused on TRPM8 only.

I do hope the comments are useful to the authors. I provide a full list of major and minor concerns:

Major concerns

I think the focus of the study is the TRPM8 with evidence of being a thermosensor. I wonder why you introduce and analyze a full family of different TRP if you focus and discuss just one.

I suggest removing UV protective function from Introduction and Discussion. It is a totally different framework (mainly related to altitudinal gradients with different exposure to UV). Also, you didn't measure or consider in the results. Therefore, it is out of context.

Rearrange sections Methods and Results. Methods need detail and explanation about why you performed the analyses and clarify the experimental design. Provide a biological relevant argument to temperatures. The style of your Results sections mixed introduction, new questions and discussions sections.

Line 66. "Thermal melanism hypothesis". Authors should be careful to consider this framework because this mechanism is more clear/important in terrestrial animals who are under the influence of direct light/energy without the buffer system (water temperature). I don't know how much differential temperature would provide light in aquatic and small (without thermal inertial) because the water temperature can be more important than the absorbance of energy driven by colouration (also, how clean is the water and how deep are the individuals would influence this relationship). I wrote this comment and after that I read you considered this on the Discussion section (knowing this framework and also because it sounds obvious you should consider clarifying the topic in the introduction).

Introduction needs a clear hypothesis. An interest paper I suggest is this: Betts et al 2021. When are hypotheses useful in ecology and evolution? *Ecology and Evolution*, 11 (11) <https://doi.org/10.1002/ece3.7365>

I think this simple article sometimes is useful to think about how to present our manuscripts.

Results section. I think this Section should be rewritten and you must add as first subsection the explanation of the experimental design. You have to provide, total sample size, sample size per treatment, origin of specimens, biological/ecological relevant argument of temperature? It's not clear why these 2 temperatures: 24 and 6 Celsius degree

Results section provides a combination of sentences from Introduction (sounds redundant sometimes) with methodology and discussion. I suggest keeping this Result section only with results. It will be so helpful to have the clarification of methodology on the proper Section. For example, lines 211 to 217; 291-293 is Introduction; lines 218 to 221, 225-231; 317-319 could be methods; lines 233-237, 279 to 283 is Discussion). Lines 294-300; 312-316; 322-324 are not results. Line 380 -395 mixed part of introduction and methods.

Figure 2B. Why didn't you measure the pigmentation index at 10 minutes, 20 minutes, 30, 40, 50 minutes in treatment 24 Celsius degrees? It looks like 2 treatments with different manipulation

and measurement patterns.

Line 234-135. This part about dark conditions were not presented in the Introduction section. It is a totally different framework to be considered and add confusion to the experiment. I didn't think of it as part of your questions in this manuscript. It is super interesting but difficult to interpret. How long were the individuals in dark conditions?

Line 243-245. I think this data could be easy to obtain.

Line 247 -249 and gradually changing temperature experiment. The variable of interest is the speed of temperature change (I agree it is interesting but was not part of the research questions). Anyway, the way you did (I accept it is creative) using different volumes recipients is not relevant. The biological variable is the speed of change and you have measured it. You can explain the method in the Method Section. Therefore, Figure 2E will be improved using temperature change speed.

Line 265. The adaptation to a dark background is again, a new question, design and potential different study. To evaluate the lines 270-278 details about methodology (how long, treatments, etc.) are required.

Lines 301-311. I don't fully understand why you decided to explore the full family of TRP channels when you have strong evidence to explore TRPM8 (based on Myers et al 2009). So, what is the question involved in analyzing all these channels? Could be part of a preliminary study.

Lines 392-395. This should have been part of the preliminary study.

Lines 399-408 and Figure 5. These are interesting results but... I wonder why the basal treatments of 24 and 0 WS12 μ M have different values. I expected similar locomotor performance between these treatments. I observed locomotion of 200 vs 600 aprox. and different Pigment index: 3500 vs 4500 aprox.

Lines 495-509. UV protection should be deleted from the Discussion section.

Minor concerns

Unify criteria to mention Trpm8, TRPM8, Trpm8, Trp (line 34, line 36, line 289, line 302 are examples of different styles) or is any reason for that?

I suggest to avoid the name PF05105679 in the Abstract

Line 39. I think it is super strong sentence and you don't provide evidence for "in ectotherm" because ectotherms include several taxa you haven't study (your line 86 suggest this potential difference)

Line 46. You should add citations for sentence in line 45 to 46. Also, maybe you can add "stable" body temperature. Because several ectotherms species that also maintain their body temperature over a range of environmental temperatures. So, you must highlight the difference with homeotherms.

Line 50. Citation (2-4) are focused on colour and these articles are useful for the next sentence. Add proper cites.

Line 54. "altering locomotor performance" is a consequence of thermal. Locomotor performance is usually influenced by temperature (as a consequence, and not as a way of thermoregulation). This sentence presents the locomotion as a behaviour used to thermoregulate. I can see that Spencer et al showed how development is influenced by temperature and then optimized the behaviours but I think this generalization is confusing.

Line 63. Delete UV protection. I think it is not relevant in this framework.

Lines 72-77. All explanations about UV harm are not proper here.

Line 93. Change connector "In contrast". In contrast to what?

Lines 102-104. This sound like a result/ conclusion

Line 111. Remove 1 bracket ")"

Line 112. Add "solution" after Ringer's

Line 124. Which concentrations?

Line 126-128. When the reader reaches this part will wonder... Which temperature (mean, medium, maximum temperature? air, substrate, water temperature? Daily, monthly, annual? Also, I wonder which would be the most relevant variable.

Line 133. It would be useful to describe the specific parameters of photos. Also, I wonder if the format was JPG or RAW (be careful because some cameras modify automatically the white balance when recording JPG files).

Line 135. I suggest improving the explanation of density of positive pixels. This means percentage of area? Did you control/measure the total surface area (ROI)? What happens with grey points ... specify threshold to consider 1 or 0.

Line 136. You mention "treatments" here but at this point you have never mentioned the treatments. It's confusing for the reader.

Line 137. Do you have some treatments with sample size = 3?

Lines 138 to 165. The audience doesn't have any clue at this stage why you are doing these analyses.

Line 149. Tail are part of the whole embryos you mention just before. Why did you separate?

Line 193. What type of locomotor performance did you measure? Speed, distance, acceleration? If it was speed, linear, curvilinear, max, mean? If it was distance, total distance? Or the trajectory during a certain time?

Line 231. I wonder if you did the reverse of this experiment. What happens with pigmentation when you increase the temperature from 6 to 24 (it would be easy and direct results, it is just an idea).

Figure 2A. I recommend adding the 16 Celsius degree phase to this diagram. Also, why did you increase from 16 to 24 Celsius degrees? I think it would be nicer to keep the base temperature at 16 and have 3 treatments (stable at 16, increase to 24 and decrease to 6). Why? Because in your experimental design you have 1 treatment with variability of temperature (24 to 6) and another without changing temperature (effect of variability vs no variability). Anyway, it is just an idea.

Line 380-382. Why do you include a different species here? Not relevant

Line 523. Any idea about not response during stage 43?

Line 528. What do you mean by long term?

Below we address the Reviewers' comments on a point-by-point basis.

Answers to Reviewer #1

We thank Reviewer#1 for valuable comments on our manuscript which improved this new version of the submitted manuscript.

- 1) The Reviewer suggests using the terms poikilotherms and homeotherms, which indicate the features of variable or maintained body temperature in animals, respectively. They suggested avoiding using the term ectotherms (exotherms).

Answer: In agreement with Reviewer #1's suggestion the term ectotherms has been changed to poikilotherms.

- 2) We make sure that the numbers of individuals used in each experiment is indicated in the relevant figure legend.

- 3) Reviewer #1 indicates that the screening of TRP channels in *Xenopus* is redundant (also see Reviewer #2 comment), in that TRPA1 is a heat-sensor. They also suggested we check the involvement of TRPM2 as a thermosensor.

Answer: We agree with Reviewer's suggestions. Accordingly, we moved the genetic screening of all *trp* channels to supplementary information, as these data will be useful to researchers employing *X. laevis* as a model organism. In the main manuscript we now focus only on the Trpm family members. As suggested by Reviewer #1, we attempted to analyze the involvement of Trpm2 and Trpm3 through pharmacological studies; JNJ-28583113 and ononetin, which are Trpm2 and Trpm3 blockers, respectively. Unfortunately, the specificity of these drugs has yet to be determined. In our hands, ononetin (a naturally occurring deoxybenzoin) killed the embryos at 100 μ M and 10 μ M, which made it impossible to determine its effect on skin pigmentation. For JNJ-2858311, the compound cost US\$ 2658 (quotation from ProbeChem), which is beyond the budget of a small grant to cover, in particular because the drug's specificity and functionality in *Xenopus* is unknown.

- 4) Reviewer #1 indicates: "it remains not entirely clear whether TRPM8 is involved in the slowing down of motor activity, it may be due simply to a decrease in muscle temperature and this process goes in parallel with the activation of TRPM8".

Answer: We agree with the Reviewer's assessment. We now raise this possibility in the Discussion (line 691): "Of note, spinal motor neurons also express Trmp8¹². While it is possible that low temperatures affect locomotor performance in a Trpm8-independent manner through alterations in the physiological properties of skeletal muscle cells, the fact that the Trmp8 agonist produces a similar slowing of embryo movement as cold suggests that Trpm8 thermosensation is involved.

- 5) Reviewer #1 indicates that skin lightening may associate with mimicry rather than thermoregulation, in that in cold water transparency is greater. Reviewer#1 pointed out the example of decreased pigmentation in many protozoa.

Answer: While Reviewer #1 is correct about the effect of temperature on water clarity of several ecosystems, is it worth noting that in general the process is slow and mainly seasonal, which differs from the rapid response in skin pigmentation we report in *Xenopus*. In general, it is not the

temperature itself that gives winter water greater clarity, but the effects that the cold water has on what lives in those ecosystems, mainly algae and other microorganisms, with duplication and the numbers of organisms increasing under warmer conditions. As pointed out by Reviewer #1, pigment responses change in many protozoa that in general are less pigmented in winter/cold water, suggesting a possible mimetic mechanism in invertebrates. Further research is necessary to understand mimetic pigmentation in invertebrates, however, is worth noting that the mechanism that triggers change in skin colour differs considerably between Chordata and non-Chordata species.

- 6) We appreciate that Reviewer #1 indicates that “a lot of work has been done with a variety of methods; an interesting result has been obtained on the participation of the cold-sensitive ion channel TRPM8 in the regulation of skin pigmentation, another functional significance of this ion channel”.

Answers to Reviewer #2:

We appreciate Reviewer #2's' comments and suggestions. Based on their recommendations we have changed this revised version of the manuscript, focusing on four of their main concerns:

A) Reviewer #2 indicated that our manuscript “**does not seem to be hypothesis-motivated**”. Answer: Our statement in paragraph four of the Introduction (previous line 100): “Whether in ectotherms Trp channels mediate more rapid, thermoregulatory responses to changes in environmental temperature is unknown”, implies that our hypothesis was that trpm channels are involved in physiological thermoregulatory responses in poikilotherms. In order to support this hypothesis, we described the current knowledge of thermoregulation and ultraviolet (UV) protection as two important means for regulating skin pigmentation (Introduction, 2nd paragraph) and the differences in thermosensation regulation by Trp channels between poikilotherms and homeotherms (Introduction, 3rd and 4th paragraph), emphasizing their role in behavioural responses. We now introduce crypsis as an additional driving force on skin pigmentation following Reviewer #2's suggestion (see point 4 below). Additionally, we have introduced text modifications in order to highlight the hypothesis of our research.

Introduction (line 120), a sentence now states: “We hypothesised that in poikilotherms Trpm channels mediate rapid, thermoregulatory responses to changes in environmental temperature”.

B) Reviewer #2 pointed out that “**There is no mention of experimental design in the Method section making it difficult to understand why you performed the analyses**”. Answer: We are somewhat puzzled by the comment, in that after >20 years in the field of experimental biology we have not encountered papers where experimental design is described in all parts of the Materials and Methods. Indeed, it is common practice for manuscripts in experimental biology that focus on mechanisms involved in behaviour and/or physiological responses to provide in the Methods section a detailed technical description of the **methods** to facilitate reproducibility by others. In contrast, the remark of Reviewer # 2 regarding “the question to be answered before any experimental design” (rationale) are stated in the Results sections before describing the experiment and the results obtained. Despite feeling comfortable with our structure

based on common practice, we have introduced several modifications to facilitate the reader's understanding of the rationale behind experiments. Hopefully, these modifications make the Methods section more "detailed oriented" and "integrated" with the Results section.

- C) Reviewer #2 suggested to improve the "Results" section by **removing redundant sentences with the Introduction section and the analysis of non-TRPM channels.**

Answer: The non-TRPM channel analysis was moved to a supplementary figure (See Reviewer#1 point 3). We keep the explanation (rationale) of each experimental design in Results, without which readers would struggle to follow the experiments. Of note, some redundant sentences were removed.

- D) Reviewer #2 suggested we remove from the Discussion "**paragraphs not related to this study (melanism as UV protective).**"

Answer: While understanding Reviewer #2's point of view, we decided to keep in the Discussion some consideration of melanism effects that may directly or indirectly be associated with the role of the Trpm8 channel.

Additional answer to Reviewer #2 (major concerns):

- 1) As suggested by Reviewer #1 and #2, the analysis of non-TRPM channel members was moved to Supplementary Figures, and we now focus only on the TRPM subfamily, with particular interest in Trpm8.
- 2) Reviewer #2 suggested we remove from the Introduction the "UV-protective" function of pigmentation.
Answer: Explanations of physiological responses that alter melanism, including temperature, UV protection, cryptic coloration and circadian variations are all mentioned in the Introduction. These explanations are important to provide readers with a platform to understand the rationale of several experiments described in the Results section. Nevertheless, we shortened the descriptions, restricting them to the context of mechanisms that alter melanism.
- 3) Our approach is a different style to that preferred by the Reviewer, but in line with that commonly used in our field. Since this is a stylistic choice, we would prefer to leave much of the organization as is. However, the Methods and Results sections were modified in order to improve clarity and the experimental design of each study.
- 4) Reviewer #2's suggested that the "melanism hypothesis" in terrestrial animals, and the "buffering" and "transparency" properties of the water influencing melanism in aquatic animals should be mentioned in the Introduction. See answer to point #2
- 5) Hypothesis is stated in the Introduction (see answers to Reviewer#2, point A).
- 6) As a stylistic preference, we have not rewritten the Results section, but have clarified methods and experimental rationale where appropriate. The explanation of the experimental design, total sample size and sample size per treatment are provided in Results section and in Figure legends. The rationale of choosing 24 and 6 degrees C for our experiments is stated in the Results section (line 257): "To identify the temperatures normally experienced by *X. laevis*, we looked to the average recorded temperatures of three national parks in southern Africa (Etosha/Namibia; Kriger/South Africa and Hawange/Zimbabwe), regions to which *Xenopus laevis* are native⁴⁵. Based on the daily

variation of temperature during the winter months (Figure 1 A and 1 B) we chose a cooling paradigm from 24 °C to 6 °C for our experiments.

- 7) We describe our approach to writing our Methods and Results sections above. For Methods we provide technical details needed to carry out the experiments. In Results, we provide a high-level explanation of the experiment; a Reader could then go to Methods if they required more detail. Further, in our Results section, to help readers follow the experimental approach, we do occasionally repeat information in the Results that was provided in the Introduction; when important for the reader to follow the thread of experimentation. Finally, we briefly discuss “results” at the end of a set of experiments if needed to set up the rationale for the next set of experiments. For example, lines 233-237 (original manuscript) are needed to explain why we hypothesized that the pineal complex was involved in the cold response, a hypothesis we went on to test. Additionally, lines 279-283 do not “discuss” the data but rather summarize the results.

We have gone through the individual examples provided by the Reviewer and made changes to remove any unnecessary redundancy, though removing many of these lines entirely from the Results would make it confusing for the Reader to understand the basic experimental design, and thus we have not done so.

- 8) Reviewer #2 asked: Figure 2B. “Why didn’t you measure the pigmentation index at 10 minutes, 20 minutes, 30, 40, 50 minutes in treatment 24 Celsius degrees? It looks like 2 treatments with different manipulation and measurement patterns”.

Answer: We did several experiments to analyze the rapid change in skin pigmentation by cooling conditions (0 to 60 minutes). All these studies (N=3) showed aggregation at 6 degrees while the pigmentation of animals kept at 24 degrees did not change. For the experiment shown in Figure 2B, we were interested in determining if the cold pigmentation response remains after long-term cooling (4, 6, 12 and 24 h). Therefore, only one short-term point was measured in un-treated individuals (1 h), while measurements were obtained at multiple “long term” time points (4 to 12 h).

- 9) Reviewer indicated that including dark conditions “add confusion to the experiment” and is not “part of your questions in this manuscript” (previously on Line 234). They also asked for “how long were the individuals in dark conditions?”

Answer: Comparing the degree of skin lightening obtained by cooling and with that of dark conditions is an important aspect of this manuscript. This comparison establishes the difference between pigmentation resulting from a “cold day” with that of a “cold night”. Moreover, this result also shows that melatonin effect on skin pigmentation is more robust than that triggered by the Trpm8 thermosensor. We added an explanation of the effect of mimicry and circadian (light/dark) induced changes in pigmentation to the Introduction (line 84). The information regarding time in dark conditions is present in the legend of Figure 2C.

- 10) Reviewer #2 points out: “Line 243-245. I think this data could be easy to obtain”.

Answer: We assume Reviewer #2 refers to data regarding changes in air and water temperature variation, with the latter acting as a buffer. Data of temperature from Southern Africa aquatic ecosystems are not easy to obtain. We have modified the sentence in order to better explain the rationale for these experiments. The paragraph (line 304) now states: “Our cooling paradigm used a sudden and defined change in water temperature that we obtained from the mean air temperature obtained from national park records. But tadpoles would normally experience gradual changes in temperature, as water acts as a buffer with

temperature change occurring slowly. To detect how the pigmentation index of tadpoles adjusts to gradual decreases of temperature we designed an experiment based on the feature that different volumes of water produce distinct temperature decay kinetics upon cooling, allowing us to slowly cool the embryos' environment at different rates. To run these experiments, tadpoles were placed in either 100 or 500 ml of MMR solution at 24 °C and then moved to 6 °C”.

- 11) Reviewer #2 indicated: “Line 247 -249 and gradually changing temperature experiment. The variable of interest is the speed of temperature change (I agree it is interesting but was not part of the research questions). Anyway, the way you did (I accept it is creative) using different volumes recipients is not relevant. The biological variable is the speed of change and you have measured it. You can explain the method in the Method Section. Therefore, Figure 2E will be improved using temperature change speed”.

Answer: The rationale of the experiment was explained above (point #10). Additionally, our results also show that pigmentation changes due to a sudden switch to a cold temperature are not a response to stress. Figure 2E shows that the final pigmentation level was similar across 3 different speeds of temperature change, arguing against the suddenness of the temperature change being a variable. Note that the importance is that the speeds differ, but the actual rates themselves were less relevant (as observed in Figure 2E insert).

- 12) Embryos temperature treatment, surface colour and dark conditions are provided in Methods (line 139 to 144): “Embryos were set in petri dishes in two identical chambers (40 cm length x 20 cm width x 25 cm height) that contained two independently-powered parallel T5 bulbs (F875 cool white fluorescent; light output, 470 lumens, colour temperature, 4000; colour rendering index, 60) residing in a 24 °C or a 6 °C incubator. Chambers contained a white surface unless specifically mentioned and the petri dishes were foiled with aluminum foil on experiments requiring dark conditions”.

- 13) The analysis of *trp* channels beside *trpm* family members was removed.

- 14) Reviewer #2 suggested that the locomotor performance of stage 42/43 tadpoles should be preliminary data.

Answer: The lack of mobility of tadpoles at stage 42/43 were results obtained during our research, rather than preliminary studies. The only place for these data is in the Results section, because they provide the rationale for why locomotor performance was assessed in older tadpoles (stage 45/46). There is no “section” in the format of the journal for “preliminary” data.

- 15) The Reviewer is correct regarding differences in the basal locomotor performance between Figure 5C (approx. 200 cm) and 5D (approx. 600 cm), as well as in the pigmentation index (Fig 5D and 5E).

Answer: Notably, the results showed in these figures correspond to two different experiments. As mentioned in the manuscript (line 543), we performed the locomotor performance analysis on tadpoles between developmental stage 45 and 46. The differences observed in the basal level of movement mainly reflect slight differences in developmental stage: Likely younger tadpoles (stage 45) were used for data collection for Figure 5C in that the larvae moved less than those used for Figure 5D that we propose are slightly older (stage 46). Additionally, locomotor performance, as well as the pigmentation level, appear to change during the day. To avoid a major influence of circadian change, we performed all the analysis over the middle of the light phase (11 am to 3 pm). Yet, this is still a 4-hour

window and could also contribute to some of the basal differences in locomotion that we observed. As related to pigmentation index, each batch of tadpoles exhibit unique levels of pigmentation, therefore basal differences are expected.

- 16) Reviewer #2 indicates: “Lines 495-509. UV protection should be deleted from the Discussion section”

Answer: Our results on tadpoles argue against the “thermal melanism hypothesis” while UV protection remain as possible mechanism for melanism regulation. In the discussion, we suggest that the thermal melanism hypothesis is important for terrestrial animals, while alternative explanations are necessary for aquatic organisms. Since UV protection may play an important role in the skin pigmentation of aquatic species, we kept the explanation in the Discussion section.

Answer to Reviewer #2 “Minor concerns”

- 1) Reviewer #2 asked about differences between capital, italic or other type of letters when referring to Trpm8 (Trpm8, TRPM8, Trpm8, *Trp8*).

Answer: In this manuscript we follow the naming conventions that distinguish between proteins and genes/mRNA, as well as differences between mammalian vs amphibians.

- 2) As suggested by Reviewer #2 we removed PF05105679 from the abstract.

- 3) Reviewer #2 referred to a “strong sentence” in the manuscript (line 39): We propose that TRPM8 serves as a cool thermosensor in poikilotherms that helps coordinate skin lightening and behavioural locomotor performance as adaptive thermoregulatory responses to cold”. They suggested we have not provided data from other ectotherms, only from *Xenopus laevis*.

Answer: Yes, our data were only in *Xenopus laevis*, but in using a model organism to discover molecular mechanisms that underlie a physiological response our hope is to discover general mechanisms that work across species. We were careful when making this statement to use the word “propose”, rather than a more direct word such as “show”, when using our data to make an inference behind what happens generally in poikilotherms. We feel this qualification (the use of “propose”) is appropriate and allows the field to then consider and test experimentally what is a suggested model.

- 4) A reference has been added

- 5) Reviewer # 2 indicated “Line 50. Citation (2-4) are focused on colour and these articles are useful for the next sentence. Add proper cites.”

Answer: While reference 2 focuses on pigmentation, references 3-4 analyze both colour change and thermoregulation, therefore we consider they are appropriate.

- 6) Reviewer #2 indicates: (previous Line 54). “Altering locomotor performance” is a consequence of thermal. Locomotor performance is usually influenced by temperature (as a consequence, and not as a way of thermoregulation)”.

Answer: We think the Reviewer’s statement is debatable. Decreasing locomotor performance (movement in general) at cold temperatures allows an organism to spend energy in other processes, including regulation of temperature or general metabolism and homeostasis. Thus, decreasing locomotor performance may contribute to thermoregulation, instead of being a simple consequence of low temperature. Nevertheless, we included in

discussion the possible effect of low temperatures on skeletal muscle activity (see also answer to Reviewer #1 point # 4).

- 7) and 8) Reviewer 2 suggested we delete UV protection (line 63) in that is not relevant to the framework.

Answer: Our explanation regarding UV protection is shortened, although maintained in the Introduction in the context of the physiological relevance of pigmentation change.

- 9) “In contrast” has been removed.

- 10) Yes, this is a result/conclusion.

- 11) Bracket removed

- 12) Done

- 13) The concentrations are indicated in the figures.

- 14) To answer all the questions Reviewer #2 asks, we added in the sentence (line 154), “see Figure 1 B”, which shows mean daily minimum and maximum temperatures from every month in Southern Africa, as well as temperatures from hot days and cold nights.

- 15) A detailed description of measurements of physiological skin pigmentation indices were described previously (see ref #41). This description included all steps for conversion of pictures to binary images.

- 16) Answered above in point 15.

- 17) Line 163. We have switched the word “treatment” to “experimental groups”.

- 18) Samples size (n) was always ≥ 8 . In general, 10 embryos were included in the analysis. The experiments (N) were repeated 3 independent times. We had included this information in the Methods and in the Figure legends.

- 19) The following sentence was added (line 166): “To identify all *trpm* members in *X. laevis* we performed a screen using the sequences from a variety of organisms with a curated protein reference sequence. Those gene sequences were used to blast the *X. laevis* genome with a cut-off expected (E) value for positive EST candidates set at 0.05.

- 20) The following sentence was moved from Results to the Methods section: “Isolated tails were analyzed as a substitute for the skin, given the absence of organs and specialized tissues within the tails relative to the whole embryo”.

- 21) For locomotor performance we measured the “total distance” swam (Methods section line 221).

- 22) Reviewer 2 asked if we measured pigmentation with a temperature increase from 6 to 24 degrees.

Answer: Because the focus of our study was Trpm channels, we only analyzed the pigmentation response to cooling (24 to 6 degrees) and not to warming (6 to 24 degrees), which likely requires different Trp channels and is beyond the scope of the current study.

- 23) The focus of this manuscript was analyzing the response to cooling. While the experimental approach suggested by Reviewer #2 is interesting it does not fit within the main scope of our study.

- 24) Reviewer 2 indicated that including *X. tropicalis* is not relevant.

Answer: Including *X. tropicalis* is relevant in that is a species related to *laevis*, but which adapted to a different thermal niche. Moreover, Trpm8 channels in *X tropicalis* are characterized and the optimum locomotor performance is known, allowing for a comparative analysis between the two related species (*tropicalis* and *laevis*). See discussion line 617 to 624.

- 25) Stage 43 embryos show a locomotor performance when are stimulated by touch but do not show spontaneous locomotor performance until stage 45/46. This must be related to the ongoing development and refinement of neural circuits that control motor activity, related to developmental stage.
- 26) Long term refers to “developmental” (previous line 528) response over days at a specific temperature, as opposed to the acute response to a temperature change. We changed “developmental” to (developmental time; days).

Reviewers' comments:

Reviewer #1 (Remarks to the Author):

I am satisfied with the corrections in the article and believe that the article can be published.

Reviewer #2 (Remarks to the Author):

Dear authors

I have carefully read the new version of the manuscript "TRPM8 thermosensation in poikilotherms mediates both skin colour and locomotor performance responses to cold temperature". I didn't find substantial changes or improvements and the manuscript is almost identical to the former version. Therefore, my opinion about the manuscript is still the same.

On the other hand, I appreciate the reply to my letter, but most of my comments/suggestions/corrections were not considered. I think you assumed my critique as focused on our different writing styles. I agree that the manuscript is your own work and I respect different styles. Anyway, my concern is about the understanding of the study (which could be influenced by the style). My intention was let you know that from my perspective (thinking in the audience of the journal) the manuscript doesn't have a proper connection between the aim (scientific questions), the methods and the results (I still think the same). I think that without a proper flow of information (a clear and transparent communication of the experimental design, doesn't matter where in the manuscript) it's difficult to follow the study and the discussion seems speculative.

I think the UV protection framework is out of context and you didn't test it. In this version, you make the hypothesis explicit and I believe as it is now, is not totally novel or related with the methodology. The sample size (totally absent in the former version) is included now but you don't specify the sample size of each treatment (I think it is necessary). Also, it's not clear how did you include the effect of repetitive measurements on statistics analysis.

Below we address the Reviewers' comments on a point-by-point basis.

Answers to Reviewer #1

The Reviewer felt we had adequately addressed their concerns.

Answers to Reviewer #2:

We appreciate the comments and suggestions of Reviewer #2. In the original revision we had made modifications to the text of the manuscript based on the concerns raised in Review 1 of the paper, Reviewer #2, however, felt these were inadequate. We apologize. Editing the text of a manuscript based on a written review is difficult, in that there is interpretation involved as to how to address issues raised. We have revisited the original review of Reviewer #2 and have revised the text of the manuscript to improve the readability of the manuscript, the flow of the experimental rationale, and the separation of text between the different sections (reduced repetition). We address our changes below:

- A) Reviewer #2 originally indicated that our manuscript “does not seem to be hypothesis-motivated”, and in Review 2 indicate that “In this version, you make the hypothesis explicit and I believe as it is now, is not totally novel or related with the methodology”

Answer: In the Introduction we have rewritten the hypothesis and provide text as to how we address the hypothesis experimentally. In so doing we make clear the novelty of the study and its relation with the methodology. Additionally, we make clear how our study, focusing only on the variable of temperature, allows us to remove other features of “light” that could influence skin pigmentation in order to understand how temperature regulates skin pigmentation in a poikilotherm model.

- B) Reviewer #2 pointed out that “There is no mention of experimental design in the Method section making it difficult to understand why you performed the analyses”.

Answer: While we have always written our papers with the Methods section providing only a detailed technical description of the methods that a reader of the Results could refer to, an approach very common in the cell and molecular biology field, we have modified the text of the Methods so that experimental rationale is provided before the technical detail is described. Additionally, we removed some information from the Results to the Methods, as suggested by the Reviewer. Ultimately, this approach of describing experimental rationale in the Methods does end up with some repetition in the

Results (hopefully minimized by wording) as we felt it was really important that we also provide the experimental rationale in the Results section for reader accessibility. Hopefully, these modifications make the Methods section more “integrated” with the Results section.

- C) Reviewer #2 suggested to improve the “Results” section by removing redundant sentences with the Introduction section and the analysis of non-TRPM channels.

Answer: In the initial revision, based on suggestions from Reviewer #1 and #2 we removed the non-TRPM channel analysis to a supplementary figure. In the current Review process, we additionally have removed some details from the Results to either the Methods or Introduction. We keep the explanation (rationale) of each experimental design in Results, without which readers would struggle to follow the experiments.

- D) Reviewer #2 suggested we remove from the Discussion **“paragraphs not related to this study (melanism as UV protective).”, and in Review #2 that “...the UV protection framework is out of context and you didn’t test it”**

Answer: Our results showing that a cool temperature lightened the skin of *Xenopus* tadpoles are surprising, in that the thermal melanism hypothesis would suggest that darkening the skin, to allow for better heat absorption from light, would be the expected response to a switch to a cool temperature. A hint as to an explanation comes from the knowledge that cool temperatures at night (dark) also generate lightened skin. Thus, melanosome aggregation/dispersion in response to temperature may be connected with associated changes in dark/light and a need for UV protection; dark in full light (warm temperature) and pale in the dark (cold temperature). We have clarified why we discuss UV protection in our Discussion. Note that we now only briefly mention UV protection in the Introduction, and only as an argument for why it is important to conduct studies where only “temperature” is altered as a variable.

Additional answers to Reviewer #2 (major concerns):

- 1) As suggested by Reviewer #1 and #2, the analysis of non-TRPM channel members was moved to Supplementary Figures, and we now focus only on the TRPM subfamily, with particular interest in Trpm8. (see above for additional changes).
- 2) Reviewer #2 suggested we remove from the Introduction the “UV-protective” function of pigmentation.

Answer: We now only briefly mention UV protection in the Introduction, and only as an argument for why it is important to conduct studies where only “temperature” is altered as a variable.

- 3) Reviewer #2 suggested that the “melanism hypothesis” in terrestrial animals, and the “buffering” and “transparency” properties of the water influencing melanism in aquatic animals should be mentioned in the Introduction.

Answer: Addressed in original revision.

- 4) Hypothesis is revised in the Introduction (see answers to Reviewer#2, point A).

We have modified both the Methods and the Results sections, clarifying methods and experimental rationale where appropriate. The explanation of the experimental design, total sample size and sample size per treatment are provided in the Results section and in the Figure legends. We have removed the rationale of choosing 24 and 6 degrees C for our experiments to the Methods section (line 122 to 126), and briefly reiterate the rationale as we introduce the cooling experiments in the Results.

- 5) Previously answered in the response to Reviewer #1 in the original revision.

- 6) On the suggestion of the Reviewer, we modified our normal approach to writing our Methods and Results sections. For Methods we now provide experimental rationale as well as the technical details needed to carry out the experiments. In Results, we provide only a brief experimental rationale and high-level explanation of the experiment, before describing the results. Further, we have modified the Results and Introduction so that there is minimal (required) repetition of information. Finally, we removed most discussion of the “results” at the end of a set of experiments, except for summarizing the data for reader accessibility.

- 7) Reviewer #2 asked: Figure 2B. “Why didn’t you measure the pigmentation index at 10 minutes, 20 minutes, 30, 40, 50 minutes in treatment 24 Celsius degrees? It looks like 2 treatments with different manipulation and measurement patterns”.

Answer: We addressed this concern in the original review, hopefully to the satisfaction of the Reviewer as they did not specifically mention this concern in Review #2.

- 8) Reviewer indicated that including dark conditions “add confusion to the experiment” and is not “part of your questions in this manuscript” (previously on Line 234). They also asked for “how long were the individuals in dark conditions?”

Answer: At night, two variables that might be important to the degree of skin pigmentation come into play; light and heat/temperature. To understand to what degree the temperature variable contributes to the lightening of the skin under dark conditions it was important to compare the degree of skin lightening obtained by cooling and with that of dark conditions. The fact that “temperature alone” (and *Trpm8* activation) causes skin lightening that is intermediate to that observed in the dark indicates that both temperature and light contribute to skin pigmentation in the dark. We have revised the rationale for these experiments so the reader understands why they are included in the manuscript. The information regarding the time in dark conditions is present in the legend of Figure 2C.

9) Reviewer #2 points out: “Line 243-245. I think this data could be easy to obtain”.

Answer: We addressed this concern in the original revision, by rewriting the rationale of the experiments, hopefully to the satisfaction of Reviewer #2 as they did not mention this point in Review #2.

10) Reviewer #2 indicated: “Line 247 -249 and gradually changing temperature experiment. The variable of interest is the speed of temperature change (I agree it is interesting but was not part of the research questions). Anyway, the way you did (I accept it is creative) using different volumes recipients is not relevant. The biological variable is the speed of change and you have measured it. You can explain the method in the Method Section. Therefore, Figure 2E will be improved using temperature change speed”.

Answer: We addressed this concern in the original revision, by rewriting the rationale of the experiments, hopefully to the satisfaction of Reviewer #2 as they did not mention this point in Review #2.

11) As per our original revision, embryos temperature treatment, surface colour and dark conditions are provided in Methods; “Embryos, drug treatment and cool response” (line 109 to 140).

12) As per our original revision, the analysis of *trp* channels beside *trpm* family members was removed.

13) Reviewer #2 suggested that the locomotor performance of stage 42/43 tadpoles should be preliminary data.

Answer: The lack of mobility of tadpoles at stage 42/43 were results obtained during our research, rather than preliminary studies. The only place for these data is in the Results section, because they provide the rationale for why locomotor performance was assessed in older tadpoles (stage 45/46). We did not move these data to Methods because Communication Biology counsels against “data not shown”, and we did not want to include “data” in the Methods section if avoidable.

14) The Reviewer is correct regarding differences in the basal locomotor performance between Figure 5C (approx. 200 cm) and 5D (approx. 600 cm), as well as in the pigmentation index (Fig 5D and 5E).

Answer: We addressed this concern in the original revision, by rewriting the rationale of the experiments, hopefully to the satisfaction of Reviewer #2 as they did not mention this point in Review #2.

15) Reviewer #2 indicates: "Lines 495-509. UV protection should be deleted from the Discussion section"

Answer: As discussed above, we have revised our discussion of UV protection to focus on possible explanations for why our data with *Xenopus* tadpoles argue against the "thermal melanism hypothesis". This point is now only briefly mentioned, but is relevant to the biological relevance of thermal regulation of skin pigmentation in aquatic species.

16) Reviewer #2 indicates "The sample size (totally absent in the former version) is included now but you don't specific sample size of each treatment (I think it is necessary). Also, it's not clear how did you include the effect of repetitive measurements on statistics analysis."

Answer: The specific sample size is included in the legend of each figure. Figures 4 and 5 now states "Each dot represents the measurement in one tadpole, and the bar is the mean with 95% confidence interval", therefore sample sizes in the figures are easily determined by counting dots. The number of experiments performed (ea. $N \geq 2$) is also included. In other figures, such as Figure 2, where each treatment is represented by the mean \pm SD, we have included in the legend the sample size (ea. $n \geq 8$) as well as the number of experiments (ea. $N \geq 3$).

Reviewer #2 also asked how we included the effect of repetitive measurements on statistics analysis. Our understanding is that analysis of "Repeated-measurements" in research applied when "subjects" are measured "two or more times on the dependent variable". In our experiments, both, the pigmentation index and the locomotor performance were measured only once in each tadpole. Moreover, in figures 5E and 5H, when we analyzed both "independent" variables in the same tadpole, single measurements were performed, therefore the analysis of repeated-measurements is not necessary.

Answer to Reviewer #2 "Minor concerns"

Minor issues were addressed in the original revision and not revisited in Review 2.

REVIEWERS' COMMENTS:

Reviewer #1 (Remarks to the Author):

I am satisfied by author's answers and recommend the article for publication.